# Learning to Look by Self-Prediction

**Matthew Koichi Grimes, Joseph Modayil, Piotr Mirowski, Dushyant Rao, Raia Hadsell**
**DeepMind**
**London, UK**
`{mkg, modayil, piotrmirowski, dushyantr, raia}@deepmind.com`

*Reviewed on OpenReview:* `https://openreview.net/forum?id=9aXKUJEKwV`

## Abstract

We present a method for learning active vision skills, to move the camera to observe a robot's sensors from informative points of view, without external rewards or labels. We do this by jointly training a visual predictor network, which predicts future returns of the sensors using pixels, and a camera control agent, which we reward using the negative error of the predictor. The agent thus moves the camera to points of view that are most predictive for a chosen sensor, which we select using a conditioning input to the agent. We observe that despite this noisy learned reward function, the learned policies avoid occlusions, and reliably frame the sensor in a specific location in the view, an emergent location which we call a behavioral fovea. We find that replacing the conventional camera with a foveal camera further increases the policies' precision.

## 1 Introduction

Computer vision, as commonly employed in embodied RL and robotics, does not closely resemble human vision. Human vision has moving eyes, fovea, movements such as saccades to frame targets in view, and smooth pursuit to track them (Dodge, 1903). In the animal kingdom, active vision in the form of background tracking is widely demonstrated across disparate branches of the tree of life, for example by goldfish, rock crabs, cuttlefish, and blowflies. This background tracking reduces motion blur, a skill without which a high-resolution retina goes to waste. It also unlocks another common ability, to detect the relative motion of small moving objects against the tracked background (Land, 1999). By contrast, in robotics and embodied RL, the camera is often rigidly fixed to the environment, limiting the agent to operate within the camera's fixed field of view (Levine et al., 2016). With the camera fixed to the environment or robot, moving objects cause large amounts of input variance as they traverse pixels. This can be a source of training instability (Cetin et al., 2022), often leading researchers to avoid learning vision altogether and use object features or off-the-shelf vision modules instead (OpenAI et al., 2019).

An agent that has learned to visually frame objects in a consistent image location could simplify the acquisition of visually-guided manipulation policies, as they can then focus on the manipulation aspect of the policy. This intuition has recently seen evidence in robotic manipulation research on hand-mounted cameras, where the object's apparent position roughly stays the same as the hand is about to grasp it (Hsu et al., 2022; Cheng et al., 2018; Gualtieri & Platt, 2018; Szot et al., 2021; Jangir et al., 2022). This "hand-chosen" camera mount consistently frames objects about to be grasped, but not necessarily other elements of the environment, such as footholds to step on, obstacles to duck, or other agents to collaborate or compete with.

By contrast, humans benefit from decoupling the kinematic chain of the eye from those of the limbs, allowing them to flexibly choose their visual focus in highly dynamic tasks ranging from fielding baseballs (McBeath et al., 1995) to traversing challenging terrain (Matthis et al., 2018). These visual policies all share common building blocks in the form of fixation and tracking (saccades and smooth pursuit). One inspiration to our paper is the question: if fixation is a general visual skill that is key to acquiring more task-specific

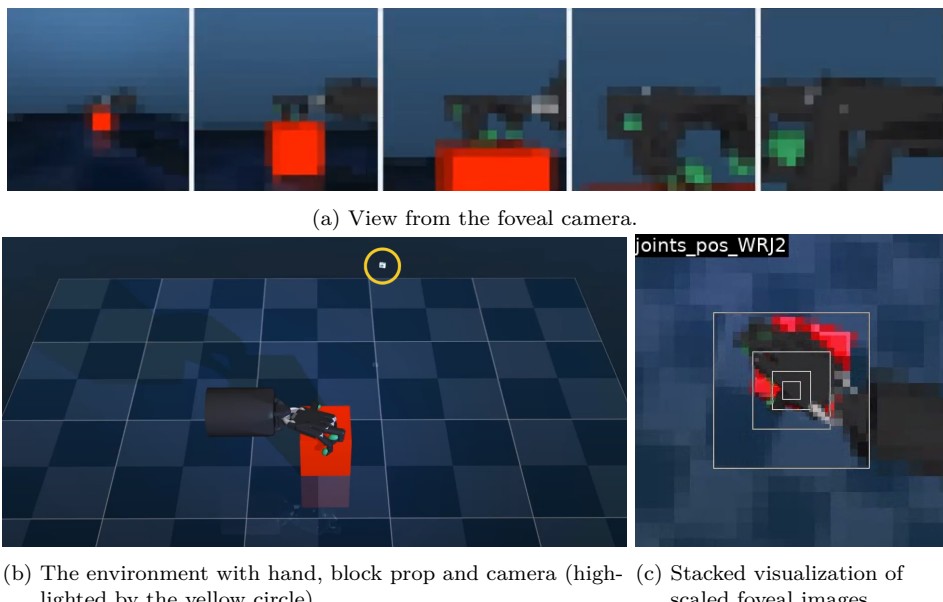

(a) View from the foveal camera.

(b) The environment with hand, block prop and camera (highlighted by the yellow circle).

(c) Stacked visualization of scaled foveal images.

Figure 1: The environment, with hand, box prop, and foveal camera. Fig. 1a shows the foveal image, a stack of 5 RGB images, with shape (5, height, width, 3). The images have the same dimension, but cover differing fields of view: 90, 45, 22, 11, and 5 degrees. Fig. 1c shows another foveal view.

visual policies, how can an agent learn it independently of specific tasks? This paper demonstrates how an embodied agent can acquire visual skills in the absence of external rewards.

We propose that learning to observe one's interactions with the world can start with learning to look at the interface to those interactions, namely the parts of one's own body. We bootstrap this by training a predictor network to predict the body's sensors from vision alone. At the same time, we use this predictor's errors as negative rewards for an RL agent that moves the camera. In other words, we reward the agent for moving the camera to viewpoints that yield better predictions for a chosen sensor. We communicate this choice of "target sensor" to the agent using a simple one-hot conditioning input. In this manner, we show that a single agent can learn distinct look-at policies, one per target sensor, without hand-designed external rewards. The predictor network, and the agent's policy and critic networks, can be trained simultaneously without manual scheduling. The learned camera policies are competent relative to baselines, navigating around occluding objects and precisely framing the targeted sensor in a consistent location within the view. We call this emergent area of the retina a "behavioral fovea".

Motivated by this emergent behavioral fovea, we implement a foveated camera (Cheung et al., 2017; Harris et al., 2019; Deza & Konkle, 2020a;b) with exponentially higher resolution near the center of the field of view. We show that this results in more precise framing behavior, as the agent learns to position the subject near the center, where it can observe it at the highest resolution.

## 2 Related Work

**Unsupervised Learning of Tracking.** Among the wider literature on tracking objects on video, some recent work focused on unsupervised learning: conditional object-centric tracking given an initial bounding box cue (Kipf et al., 2021), segmentation using Object Discovery and Representation Networks (Hénaff et al., 2022) and segmentation using motion-based (optical flow) and appearance-based information (Choudhury et al., 2022). Our unsupervised approach learns to execute large viewpoint changes in 3D environments, rather than tracking or annotating within the fixed view given by a video.

**Computational Models of Foveation.** Cheung et al. (2017) introduce a neural attention model with a learnable retinal sampling lattice, with multiple extensions (Harris et al., 2019; Deza & Konkle, 2020a;b).

Other work uses spatial attention (Kosiorek et al., 2017) or a virtual fovea (Burt et al., 2021) to imitate the equivalent of *what* (bottom-up saliency) and *where* (top-down attention) pathways in animal vision. Rivkind et al. (2022) introduce a low-resolution dynamical sensor that moves with drift-like tiny steps mimicking the microsaccades of a human eye. By contrast, we implement visual attention as a motor policy capable of driving the camera to informative viewpoints in 3D, rather than controlling which pixels to attend to within a given 2D image. Instead of cropping or masking out the non-foveal part of the image, we retain the low-resolution wide-angle periphery to aid the camera agent in navigating through a 3D environment.

**Visual Attention in Reinforcement Learning.** Attention mechanisms for agents playing RL games were introduced by Sorokin et al. (2015). They observed that top-down attention mechanisms forced agents to focus on task-relevant information by sequentially querying the environment (Mott et al., 2019) and helped generate virtual goals to replay (Liu et al., 2020). Guo et al. (2021) studied analogies between the visual attention of human experts and saliency maps in RL agents. Recently, self-supervised attention in RL agents has been used to select regions of interest without explicit annotations (Wu et al., 2021) and has provided robustness as well as increased learning efficiency and interpretability for visual tasks (Salter et al., 2020; James & Davison, 2022; James et al., 2022; Tang et al., 2020). Our active vision approach uses RL to control the camera, instead of controlling top-down attention over fixed views.

**Active Vision in Reinforcement Learning.** Embodied perception in a navigating agent enables it to move around an object to perceive it better (Yang et al., 2019) or to solve semantic segmentation tasks (Chaplot et al., 2020; Nilsson et al., 2021). In a panoptic camera rig, RL can be used to select the best viewpoint for human pose estimation (Gärtner et al., 2020), 3D reconstruction (Pirinen et al., 2019), forecasting the effects of motion (Jayaraman & Grauman, 2016), and more generally, learning to look around to efficiently gather information about the agent's surroundings (Ramakrishnan et al., 2019; Jayaraman & Grauman, 2018). Our work focuses on learning active vision skills without external task rewards.

## 3 Methods

Our method teaches a camera-controlling agent to visually frame parts of its own body. It does this by simultaneously training three networks: the agent's policy and critic networks, and a separate predictor network that predicts the body's sensor values from the camera's pixels. We use the predictor's errors as reinforcement learning penalties for the camera agent, incentivizing it to move the camera to more informative views. Below, we describe each of these components.

### 3.1 Environment

Our environment uses the MuJoCo (Todorov et al., 2012) physics simulator, in which we place a camera at the end of an invisible armature (the *camera bot*), controlled by the camera agent. This camera bot shares the scene with a manipulator, and a block prop randomly positioned within reach of the fingers, which gives the touch sensors at the fingertips something to touch (fig. 1b). Both the camera bot and manipulator are driven by velocity control, i.e. proportional-integral-derivative (PID) control in which the actions specify a target velocity for each actuator.

We treat the camera bot and manipulator as belonging to one robot that is conceptually split into two separate entities. The manipulator runs a fixed random behavioral policy throughout the experiment, but sensors on the manipulator are made available to define the losses of the predictor network (section 3.3), a multi-headed generalized value function (GVF) whose losses define the endogenous penalty function for the camera agent (section 3.4). The camera agent has to learn to move to look at selected parts of the manipulator to better predict the future values of the targeted sensor in the manipulator. The training process only modifies the behavior of the camera bot, not the manipulator.

At the start of each episode, we randomize the joint angles of the manipulator and camera bot, and the position and dimensions of the box prop. We also randomly choose a sensor on the manipulator for the camera agent to target. We use an episode length sufficient to allow the camera enough time to reach an arbitrary final pose from its initial randomized pose.

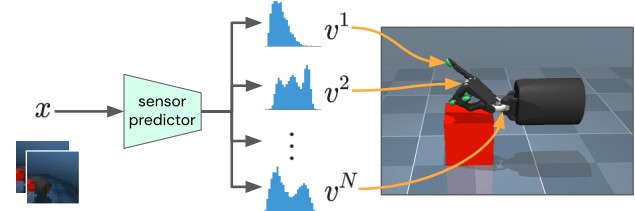

Figure 2: Predictor network input and outputs. The input $x$ is a stack of the previous two camera observations (to enable predicting velocities). The outputs $\{v_i\}_{i=1}^N$ are the distributions of the estimated return values of $N$ proprioceptive sensors, such as joint angles or touch sensor readings. For clarity, we have lightened the background on this high-resolution image rendered in MuJoCo.

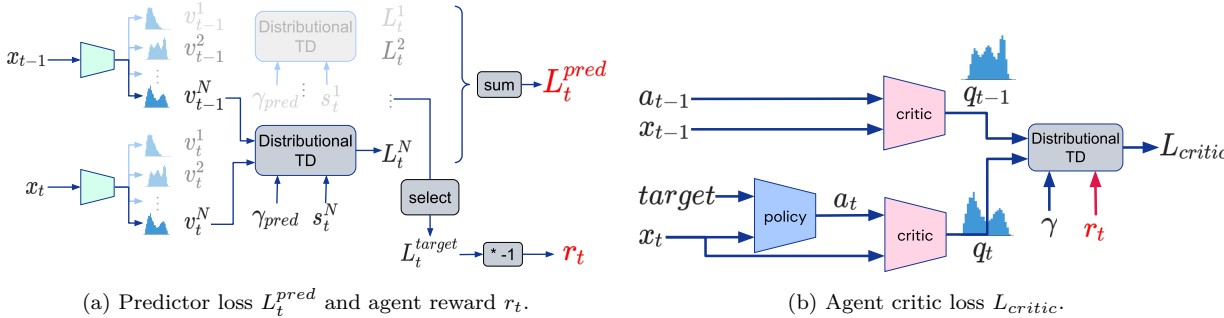

(a) Predictor loss $L_t^{pred}$ and agent reward $r_t$.

(b) Agent critic loss $L_{critic}$.

Figure 3: a) Distributional TD losses $L_t^i$ are computed for each sensor reading $s_t^i$, in a loop over $N$ sensors (the figure only shows the $N$'th sensor's reading $s_t^N$). The predictor loss $L_t^{pred}$ is the sum of the $N$ distributional TD losses. A different *target* sensor is chosen at the start of each episode, whose prediction error $L_t^{target}$ is used as a negative agent reward $r_t$. b) The D4PG (Barth-Maron et al., 2018) critic loss $L_{critic}$ depends on the learned agent reward $r_t$. The *target* sensor (see Section 3.4) is specified as a one-hot vector, fed to the policy as a conditioning input.

## 3.2   Cameras

The camera bot can be equipped with a conventional camera or a foveal one. Like Mnih et al. (2014), we implement the foveal camera as $N$ conventional cameras, all with the same position, orientation, and pixel dimensions, but differing in their field of view. Fig. 1a shows an example of the resulting multi-scale images as seen by the agent, and fig. 1c demonstrates the scales' relative fields of view by stacking them on top of each other.

## 3.3   Predictor network

The predictor network shares no weights with the policy or critic networks in the agent. It is a convolutional residual network (ResNet) followed by a multi-layer perceptron (MLP). Its architecture and layer sizes are taken from IMPALA's (Espeholt et al., 2018) convolutional residual network, replacing the LSTM at the end with an MLP. This predictor network takes two consecutive images as input, and outputs predictions for all potential target sensors. Like the critic network, its predictions take the form of discrete distributions $v_t$ of estimated future-discounted returns $y_t$ of sensor signals $s_t$ rather than of rewards.

$$y_t = \sum_{t=0}^{\infty} \gamma^i s_{t+i+1} \tag{1}$$

$$v_t(y) = p(y_t = y) \tag{2}$$

This was inspired by multi-timescale nexting (Modayil et al., 2014), and the predictor network can be thought of as a multi-headed GVF (Sutton et al., 2011) with distributional outputs. Predicting a decaying sum of sensor readings, rather than a future sensor reading at a particular point in time, makes our experiments less dependent on the exact choice of frame rate or prediction timescale. Furthermore, using distributional TD losses instead of L2 losses serves as a principled means of normalizing the prediction losses across multiple sensor modalities with very different numerical ranges and distributions.

Driving behavior to maximize prediction errors has been suggested as a form of curiosity (White et al., 2014). Our agent does the opposite of this, as it is rewarded for minimizing error. Instead of maximizing short-term surprise, it explores the state space through the diversity of its prediction target sensors. This is partially a reflection of our different setting. When an agent physically interacts with the world under a fixed camera, it may make sense to explore by maximizing prediction error. When the agent moves the camera itself, it can trivially increase prediction error by staring in uninformative directions, without meaningful exploration.

### 3.3.1 Agent networks

The agent consists of a policy network and a critic network. Like the predictor network, each of these is an MLP stacked on an IMPALA-style convolutional ResNet. The critic and policy networks share a ResNet, but have different MLP heads. Both networks take a mix of images and other inputs. The images are fed through the ResNet, and the other inputs are concatenated with the ResNet's output and fed to the MLP. The policy network takes as input the observation (two consecutive image frames), and a one-hot vector specifying the target. It outputs bounded continuous target velocities for the camera bot's four joints (section 3.1). The state-action critic additionally takes the action, and outputs a discrete distribution over the estimated return. The agent's training objective is to minimize the predictor network's error for the target sensor, by modifying the policy network to move the camera to look at the associated body part.

### 3.4 Training

For each batch of transitions sampled from the replay buffer, we compute losses for the predictor, agent critic, and agent policy networks, then perform gradient updates on all three.

To train the predictor, we compute the prediction error of sensor $i$ using the distributional TD loss (Bellemare et al., 2017) from D4PG (Barth-Maron et al., 2018). This is analogous to the standard TD error ($\delta = r_t + \gamma v_t - v_{t-1}(x)$), except that values $v$ are represented not by scalars, but by discrete distributions over the range of possible return values (eq. 2).

$$L_t^i = DistributionalTD(s_t^i, \gamma_{pred}, v_t^i, v_{t-1}^i). \tag{3}$$

The predictor discount $\gamma_{pred}$ is separate from the discount $\gamma$ used for training the critic. Setting $\gamma_{pred} = 0$ amounts to performing next-frame prediction, while setting it to larger values predicts its future sum over a decaying time window with half-life $h = \Delta t \frac{ln(0.5)}{ln(\gamma_{pred})}$. It is possible to predict over multiple time windows as in Horde (Sutton et al., 2011), which may be useful in environments with predictable dynamics over multiple frames. For our environment, where the camera and manipulator have little momentum, we use decay $\gamma_{pred}$ chosen to have a short half-life of 0.1s. The predictor network's loss is then the sum of prediction losses across all target sensors, $L_t^{pred} = \sum_i L_t^i$.

The camera agent networks share no parameters with the predictor. Each episode randomly chooses a proprioceptive sensor on the manipulator, to serve as the camera agent's *target* for that episode. The camera agent's task is to position the camera in a manner that reduces the predictor's error for that target sensor. We therefore define the reward to be $r_t = -L_t^{target}$, where $L_t^{target}$ is the prediction error for that episode's target sensor, on timestep $t$. This is a dense reward, as it is available on every timestep, but a noisy one, as it is learned. The agent critic loss is analogous to eq. 3, substituting $r_t$ for sensor reading $s_t^i$, $\gamma$ for $\gamma_{pred}$, and critic outputs $q_t$ and $q_{t-1}$ for predictor outputs $v_t$ and $v_{t-1}$. We train the policy network using the deterministic policy gradient loss (Silver et al., 2014).

## 4 Experimental Results

We describe our experimental setup, then present the claims supported by our results. All results are from the trained camera agent, evaluated without the exploration noise used during $\epsilon$-greedy training.

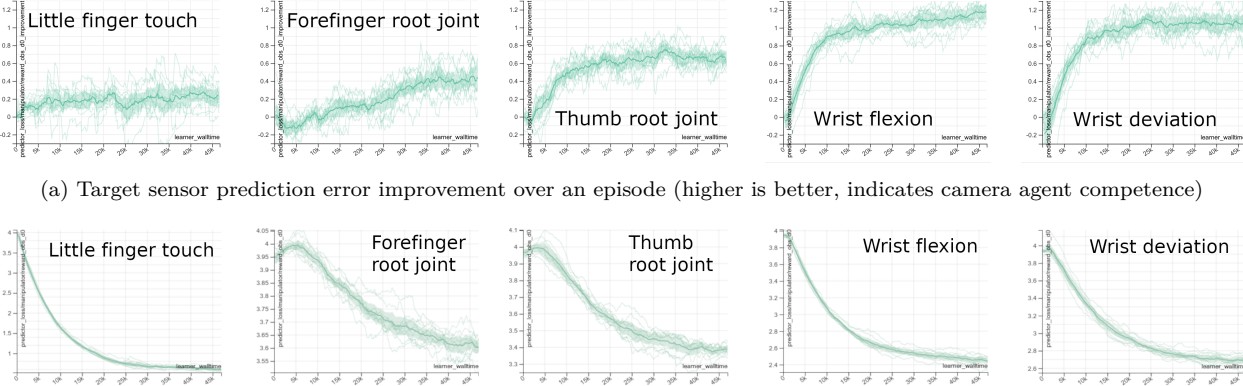

(a) Target sensor prediction error improvement over an episode (higher is better, indicates camera agent competence)

(b) Target sensor prediction error average over an episode (lower is better, indicates predictor competence)

Figure 4: **Agent and predictor training.** X axis is wall time in seconds, spanning 13 hours of training. In fig. 4a, the Y axis is the improvement of the target sensor's prediction error ($L^{target}$, eq. 3) from the beginning to the end of the episode, as the agent moves the camera to a better point of view. In fig. 4b, the Y axis is the average value of $L^{target}$ over the course of an episode. Unlike in fig. 4a, the Y axes have different ranges, to better show the progression of predictor training. $L^{target}$ is a KL divergence between distributions over possible return values so is unitless, and its range is independent of the sensor value range. All curves are taken from a single agent conditioned to target one of the above five target sensors, chosen randomly on each episode. Bold curve: the mean of 10 experiments with different RNG seeds. Shaded area: $\pm 1.96\sigma$.

## 4.1 Experimental setup

**Environment** The manipulator is a model of the 20 degrees of freedom (DOF) tendon-driven hand by Shadow Robotics (Shi et al., 2011; Plappert et al., 2018). We drive the manipulator using Perlin noise (Perlin, 1985), which provides temporally smooth control by interpolating random keypoint velocities with splines. The keypoint velocities are spaced 1 second apart and are uniformly sampled from the range of joint velocities. Noise drives all the tendons, causing the hand to writhe about. The camera bot has four DOF with different ranges: elevation $[0, \pi]$, azimuth $[-\pi, \pi]$, distance $[.2m, 2m]$, and yaw $[-\pi, \pi]$. The first three DOFs move the camera in spherical coordinates centered around the manipulator. The yaw DOF rotates the camera around its local vertical axis, allowing it to look away from the hand. The values for these DOFs are randomized at the beginning of each episode. Randomizing the yaw DOF causes the manipulator to be entirely outside the field of view half the time. The episodes are 12 seconds long. The camera agent selects actions every 200 ms, yielding episodes of 60 timesteps.

**Cameras** In our experiments, the foveal camera has $N = 5$ cameras with $21 \times 21$ pixels each. Camera 1 has a FOV of 90 degrees in the vertical and horizontal directions, camera 2 has a FOV of 45 degrees, and so on, down to camera 5 with a FOV of 5.26 degrees ($90 \times 2^{-4}$). The conventional camera is $21 \times 21$ pixels, with a FOV of $90 \times 90$ degrees.

**Training** We train the agent and predictor networks simultaneously. We employ 512 processes that each run an independent copy of the environment, pushing experience transitions (labeled with their episode's sensor targets) into the replay buffer. The buffer runs in its own process, and has a maximum capacity of 1M transitions (168 GB of RAM when using for $5 \times 21 \times 21$ foveal images). A single learner process samples mini-batches of 256 transitions from the replay buffer. The policy network trains with a learning rate of $10^{-5}$, while the critic and predictor networks use a learning rate of $10^{-4}$. We use Adam (Kingma & Ba, 2015) to minimize the losses. Each environment process runs on a machine with 1 CPU and 1.1 GB of RAM. The learner runs on a machine with 2 CPUs, 4.9 GB of RAM, and a TPU. Training takes roughly 72 hours to converge for both foveal and conventional agents.

**Networks** The agent's policy and critic networks share a convolutional ResNet, which acts as a visual feature extractor. Its architecture and layer sizes are taken from IMPALA's (Espeholt et al., 2018). The policy and

critic have separate MLP heads, with hidden layer sizes $(256, 256, 256)$ and $(512, 512, 256)$, respectively. The predictor uses the same layer sizes as the critic, but has its own convolutional network weights.

**Inputs and outputs** The predictor takes a pair of successive frames from the camera, and outputs N discrete distributions for N possible target sensors, over the range of possible return values. In our experiments, $N = 5$. These five sensors were selected to represent both touch and joint angle sensors. We selected the predicted joints to cover a range of sizes and directions of movement when the joints bend. The critic takes an N-dimensional one-hot vector in addition to the pair of pixel frames. This vector indicates which sensor to target. The critic outputs a distribution over the possible return values of the learned reward. The predictor and critic's output distributions are 51-atom discrete distributions evenly spanning the range of possible returns. The policy network takes the target one-hot and a pair of pixel frames, and outputs continuous target velocities for the camera bot's four DOFs: azimuth, elevation, distance, and yaw.

### 4.2 The trained camera policy improves target prediction accuracy

Figure 4a shows the improvement of the target sensor's prediction error from the start to the end of an episode, plotted throughout training. We plot the error improvement rather than the final error (seen in figure 4b and table 1), because the latter would jointly evaluate the predictor and the camera agent. By contrast, the error improvement within an episode evaluates the camera policy, independent of the predictor. Regardless of predictor quality, a policy selecting velocities from a uniform distribution centered at zero does not, on average, improve prediction accuracy over the course of an episode. A camera policy trained to improve this prediction error will do so, as shown. The wrist joint angles (two rightmost plots) have the most visual impact, as they move the whole hand. The agent learns to improve those errors first, while policies targeting the finer sensors improve later.

|  | Little finger touch | Forefinger root joint angle | Thumb root joint angle | Wrist flexion angle | Wrist deviation angle |
|---|---|---|---|---|---|
| Blind (c) | $0.680 \pm 0.020$ | $4.12 \pm 0.065$ | $4.11 \pm 0.073$ | $3.84 \pm 0.059$ | $4.08 \pm 0.051$ |
| Random (c) | $0.667 \pm 0.067$ | $4.16 \pm 0.034$ | $4.16 \pm 0.056$ | $3.51 \pm 0.069$ | $3.61 \pm 0.050$ |
| Random (f) | $0.648 \pm 0.048$ | $4.07 \pm 0.033$ | $4.09 \pm 0.068$ | $3.41 \pm 0.054$ | $3.58 \pm 0.077$ |
| Geometric (c) | $0.620 \pm 0.001$ | $4.12 \pm 0.033$ | $4.10 \pm 0.026$ | $3.57 \pm 0.038$ | $3.79 \pm 0.027$ |
| Geometric (f) | $0.566 \pm 0.046$ | $3.75 \pm 0.037$ | $3.61 \pm 0.044$ | $2.55 \pm 0.043$ | $2.84 \pm 0.057$ |
| Ours (c) | $0.582 \pm 0.062$ | $3.24 \pm 0.070$ | $2.98 \pm 0.061$ | $1.73 \pm 0.054$ | $2.17 \pm 0.048$ |
| Ours (f) | $0.606 \pm 0.046$ | $3.11 \pm 0.050$ | $2.75 \pm 0.071$ | $1.65 \pm 0.047$ | $1.93 \pm 0.051$ |
| Oracle (c) | $0.480 \pm 0.020$ | $2.80 \pm 0.096$ | $2.53 \pm 0.025$ | $1.45 \pm 0.017$ | $1.62 \pm 0.026$ |
| Oracle (f) | $0.587 \pm 0.022$ | $2.80 \pm 0.030$ | $2.53 \pm 0.032$ | $1.45 \pm 0.009$ | $1.66 \pm 0.010$ |

Table 1: **Target sensor's prediction error at episode end (lower is better).** The "(c)" and "(f)" indicate conventional or foveal camera. *Blind* and *Oracle* give upper and lower bounds to the error, *Random* shows the prediction error of a randomly posed camera. *Geometric* shows the prediction error of a camera with random position, but pointed at the hand, whose position is inferred by known body geometry. See section 4.2.1 for more detail. Error is measured as the TD error for predictions given as distributions over the range of possible return values. Confidence bounds indicate $\pm 1.96\sigma$, calculated from 10 runs with different RNG seeds.

#### 4.2.1 Comparison with baselines

Table 1 shows the target's prediction error at the end of the episode (lower is better). Unlike the relative measure of the agent plotted in fig. 4a, this is an absolute measure of the joint quality of the predictor and agent at the end of training. It compares trained agents against the following baselines:

**Blind:** The *Blind* baseline is a predictor trained on a camera pointed away from the hand. It can do no better than learn to output each sensor's prior distribution, and serves as an upper bound to the expected agent prediction error.

**Oracle**: For a lower bound on the prediction error, we run a sweep over a series of hand-chosen fixed camera poses surrounding and looking at the hand, training a separate predictor for each pose. The *Oracle* entries show the minimum prediction error over all viewpoints for that target sensor. The Oracle benefits from not only specializing to a single point of view, but also from not moving, which significantly reduces data variance and improves prediction error even in the moving-camera agent. In practice, moving agents cannot

Figure 5: Agent performance under sensor noise. Vertical axis: prediction improvement of the target sensor over the course of an episode, due to the camera agent repositioning the camera to a better view. Horizontal axis: Number of actor episodes. Colors represent different amounts of noise added to the sensor readings. The noise is parameterized by $\alpha$, the fraction of the sensor range covered by $\pm 1$ standard deviation of the noise distribution. See section 4.3 for details.

always maintain a static view, nor can an agent with multiple static cameras usually know a priori which camera will yield the most accurate predictions.

**Random**: The *Random* agent is our agent with its policy and critic learning rates set to zero. The camera spawns randomly, as usual, but hardly moves thereafter (brownian actions do little to move the camera bot, which has high inertia). The predictor must learn to predict from the resulting random camera views. These views are mostly static, advantageously reducing input variance in a similar manner to the still images of the Oracle. Our agent outperforms the Blind and Random baselines by a statistically significant margin.

**Geometric**: The *Geometric* agent is identical to *Random*, except instead of randomizing the camera's position and orientation, *Geometric* only randomizes the position. The orientation is pointed at the hand, using the known relative geometry between it and the camera. The fact that the Geometric baseline policy performs worse than the learning policy shows that using externally specified geometric knowledge of the sensor location is suboptimal in this setting. This serves as an upper bound on what a classical control policy of orientation could achieve. The foveal version gets lower errors than its conventional counterpart, as its multi-scale camera allows it to see detail even when randomly positioned at a distance.

## 4.3 The method is robust to epistemic and aleatoric uncertainty

As our method uses prediction accuracy as a reward, here ask how it fares in the face of uncertain predictions. Broadly, predictions may be uncertain due to external, aleatoric sources of uncertainty, such as sensor noise, or internal, epistemic uncertainty, such as a sub-optimal predictor.

**Aleatoric uncertainty** Simulating sensor noise accurately is an active field of work, much of it idiosyncratic to each sensor type or even instance. However, we may still use simple noise models to make general probes into our method's response to sensor noise. In fig. 5, we plot the target sensor's prediction error improvement over the course of an episode vs actor steps. In section 4.2, we use this improvement as a measure of the camera agent's competence. We run a sweep of five levels of noise, with three random seeds each, for a total of 15 experiments. Here we used a conventional camera with $47 \times 47$ resolution. We added noise to all non-visual sensors, sampled from zero-centered normal distributions with variances parameterized by a hyper-parameter $\alpha$ as follows. The standard deviation of each sensor's noise is set to $\sigma = \alpha(s_{max} - s_{min})/2$, where $s_{max}$ and $s_{min}$ are the sensor bounds. For example, if $\alpha = 1$, then the noise's $\pm 1$ standard deviations would span the whole sensor range. After adding the noise, we clamped the sensor value to its range. Fig. 5 shows the performance curves under $\alpha = 0, 0.05, 0.1, 0.2$, and $0.4$. All plots show that while increasing noise diminishes performance as expected, the camera still learns competent policies without being derailed, even under sensor noise variances that would be considered severe in a robotic context.

**Epistemic uncertainty** The predictor, which learns the reward function, and the camera policy, which learns from this reward, are trained from scratch in parallel with no effort made to schedule the reward training to precede the camera agent training. The agent nonetheless learns a competent policy, indicating robustness to the high epistemic uncertainty exhibited near the beginning of training by the untrained predictor. Fig. 4a shows the camera agent's competence, as measured by the improvement of the target sensor prediction over an episode due to camera motion. Below it, fig. 4b shows the predictor competence, as measured by the average prediction error of the target sensor over the episode. In all instances, the agent

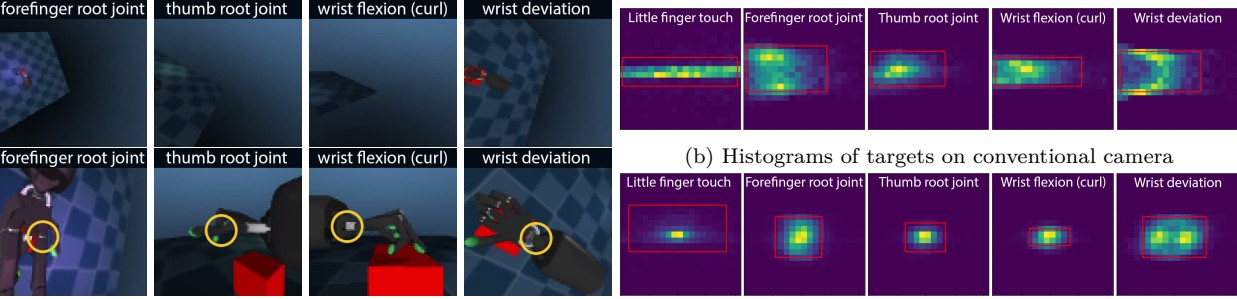

(a) First (top row) and last (bottom row) frames

(b) Histograms of targets on conventional camera

(c) Histograms of targets on foveal camera

Figure 6: Fig. **a**: the first and last frames (top and bottom row) from episodes where the sensor in the caption is the visual target. The figure highlights the target sensor with yellow circles (the agent does not see this). Note that the agent chooses to observe the wrist flexion angle (3rd column) from the side, from where it is most visible, while it observes the wrist's side-to-side deviation angle (4th column) from above. Fig. **b** and **c** show histograms of target location at the end of the episode, in image space, for conventional and foveal agents. The red box covers the 95% confidence interval (2.5% to 97.5% of the cumulative probability distribution) along each axis. The histograms accumulate the final target position over episodes collected after convergence, or the last 8 days out of an 11 day training run. This amounts to a total of 25000 episodes, or roughly 5000 episodes per target.

performance starts to plateau well before the predictor performance has converged. For example, when the target is the wrist deviation, the camera agent curve enters its plateau stage at 10k seconds of training, at which point the corresponding predictor is only 51% of its way from its peak value (4.0) to its value at convergence (2.7).

## 4.4   The trained camera policy frames the subject

A consistent outcome in our experiments is that camera agents trained on a fixed choice of target tend to place the target in a particular position on the screen. This position varies from one training run to another, though it is usually near the center (fig. 6a). This is not an instance of the camera agent having memorized a particular set of preferred values for its own joint angles. The camera agent is unaware of the camera bot's joint angles, as the only inputs are pixels and the one-hot vector specifying the target sensor. Furthermore, it controls the camera bot joints by velocity control, unlike a position-controlled camera, which may learn to output a constant target camera pose regardless of the input. Figure 6b shows the emergence of this *behavioral fovea* from framing.

## 4.5   Training the camera policy with a foveal camera yields more precise framing

As shown in fig. 6b, agents learn to frame the target in a specific region of the image, even when equipped with a non-foveal camera. We call these regions *behavioral fovea*. Equipping agents with a foveal camera induces more focused and centered framing behavior (fig. 6c), as the agent can see a target at full resolution only when framing it at the center. A stronger tendency to center the subject further lessens the need to spend network capacity on position-equivariance. If the target moves around the workspace, tracking behavior may emerge as a side-effect of this tendency to keep it centered in the image. We discuss this future work in section 5.

The non-foveal camera places the behavioral fovea near the center despite having no physical fovea there. Therefore, one might ask whether the foveal camera has a tightly centered behavioral fovea primarily because that is the location of its physical fovea, or whether this is mostly an enhancement of a preexisting central bias in the learning dynamics. Figure 7 addresses this question. Each row shows the behavioral fovea for a single experiment, with the physical fovea horizontally shifted to the left, center, and right. In each case, the agent learns to position the target sensor on the physical fovea, showing it to be the determining factor in where the agent chooses to frame the target.

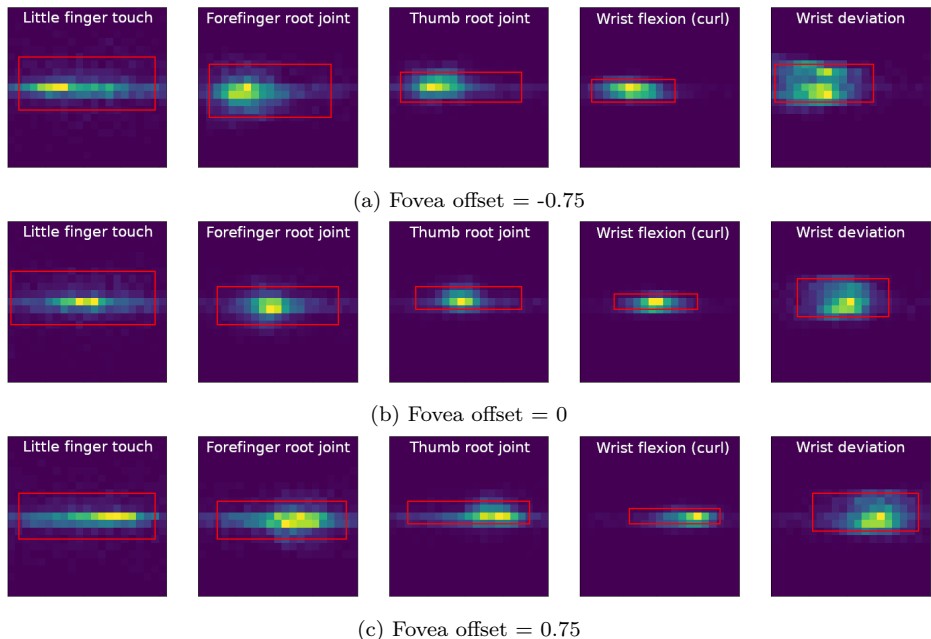

Figure 7: Histograms of the target sensor location at the end of the episode, in image space, for foveal cameras with an off-center fovea. Each row is labeled by the horizontal offset of the fovea, in normalized image coordinates (-1 and 1 correspond to the left and right edges of the image). Each column corresponds to a different target sensor. The red box covers the 95% confidence interval along each axis.

Figure 6 also helps answer a fundamental question: why use RL to learn to look at sensors, as opposed to supervised policy training? After all, the positions of sensors are calculable from the robot's 3D geometry, which could provide image-space targets to direct the camera towards during training. Our RL-based method has the following advantages over such a supervised approach: (1) it needs no such prior information on the robot's geometry, (2) the agent learns not just what 3D point to look at, but from which direction, and (3) the most informative point of focus is not always the sensor itself. Fig. 6a illustrates point 2. The agent observes wrist flexion and deviation from orthogonal directions (i.e., along their orthogonal bending axes), despite looking at the same wrist. The "geometric" baseline in section 4.2.1 perfectly orients the camera at the hand, and thus represents an upper bound on the performance of an agent supervised on orientation alone. As seen in table 1, it underperforms our agent, which learns a suitable orientation and position for the camera. The rightmost image of fig. 6c illustrates point 3. The histogram shows two peaks; the wrist is framed off-center to the left or right, and the camera centers the hand instead. This is because the wrist primarily causes motion to the hand, making the latter more informative to look at. (Some sensors have a larger decoupling between the sensor position and the most informative view, such as an IMU-based orientation sensor attached to some arbitrary location in a large rigid body.)

## 4.6 The camera policy adopts distinct camera positions for different sensor targets

Sections 4.4 and 4.5 showed that the policy learns to orient the camera to frame the target sensor at a specific image location. While it is obviously important to look in the right direction, looking from the right position also matters. Fig. 8 shows the distribution of the camera position at the end of the episode for a single trained agent, showing a separate plot for each sensor target. Figure 8a shows that the camera has learned to observe the wrist flexion in profile, i.e., along the axis of rotation, from which the visual flow of flexing the wrist is most apparent. It observes the wrist from one side or the other, hence the bi-modal distribution seen in the red projection. By contrast, observing the same wrist joint, but predicting its side-to-side deviation angle, behooves the agent to adopt a top-down view, as shown in fig. 8b. The differences in distribution for the camera positions when observing the finger root joints (fig. 8c, fig. 8d) are more subtle, observable in the blue projection along the floor. Fig. 8e shows the agent to be more position-agnostic when targeting the

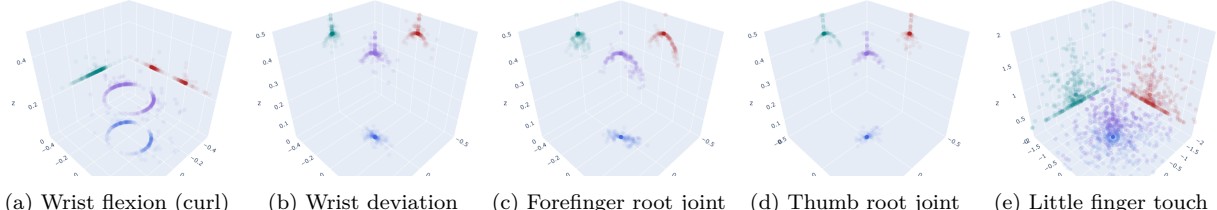

(a) Wrist flexion (curl) (b) Wrist deviation (c) Forefinger root joint (d) Thumb root joint (e) Little finger touch

Figure 8: Positions of the camera at the end of the episode, for different target sensors. Positions are in purple, with their projections to axis-aligned planes shown in red, green, and blue.

little finger's touch sensor. As shown in fig. 4a, the little finger's touch sensor is the most difficult to predict, due to the severe label imbalance presented to the predictor (the touch sensor touches the block infrequently under the hand's random policy). This leads to a noisy learned reward, which is good enough to teach the agent to center the touch sensor in the view (fig. 6c), but is insufficient to narrow down the camera position. In section 6 we propose a method to improve this.

### 4.7 The camera policy learns to circumnavigate occlusions

We introduced a randomized occluder to our training environment to test the generality of our self-supervised training. The occluder is a flat rectangle with random color, position, and dimensions. We uniformly sampled its height and width from the range $[0.5, 1.0]$, and sampled its position in spherical coordinates from $azimuth \sim [-\pi, \pi]$, $elevation \sim [0, \pi]$, $distance \sim [.4, .6]$. These spherical coordinates are roughly centered around the hand. For reference, the size of the environment's floor is $2 \times 2$. To maximize its effectiveness as an occluder, we orient the board to face the hand. Fig. 9 shows two trajectories of the camera agent, starting from the same initial conditions. The resulting camera policy is able to sidestep the occluder when it blocks the camera's view, thereafter exhibiting similar viewpoint preferences as in the unoccluded case.

## 5 Discussion

Many animals do not have a fovea. Some, like rats, have a roughly even distribution of optical receptors spread over a nearly spherical field of view. When objects in all directions are seen with equal acuity, one might ask if rats need to move their eyes to look *at* subjects at all. Yet they do, exhibiting similar visual skills to humans, such as centering and fixating the subject in a specific area of the retina (Holmgren et al., 2021). As with the rat, we find that our non-foveal agents exhibit framing behavior, positioning the subject in a specific area of the image, giving rise to a *behavioral fovea* despite there being no intrinsic acuity advantage in one area of the image versus another.

This suggests that the dynamics of the training encourage a positive feedback loop between the predictor and agent: the predictor improves its expertise in a specific region of the field of vision, and the agent learns to move the relevant subject into this visual region to receive the better prediction reward. This in turn provides to the predictor even more training data with the target in that image position, further improving its predictions there. This behavior is in contrast to the usual emphasis placed on learning position-invariant or position-equivariant representations in computer vision research that uses static datasets of images rather than an active camera (LeCun et al., 1998). That said, even static facial recognition has been shown to benefit from normalizing the facial feature locations (Taigman et al., 2014), a domain-specific form of framing.

**Limitations** We simulated the Shadow hand in the MuJoCo environment and thus have yet to investigate real-world complexities such as sensor timing, synchronization, noise, resolution, and discretization. Predicting distributions of returns instead of sensor readings isolates our framework from some of these real-world complexities but not all (e.g. motion blur). We use a random policy for the hand, under which the fingertips rarely touch the block, resulting in highly skewed distributions for their touch sensors. This results in relatively poor touch prediction compared to joint angle prediction. In future work, we plan to reward the hand for maximizing target sensor entropy, yielding more pedagogical hand behavior with a flatter sensor

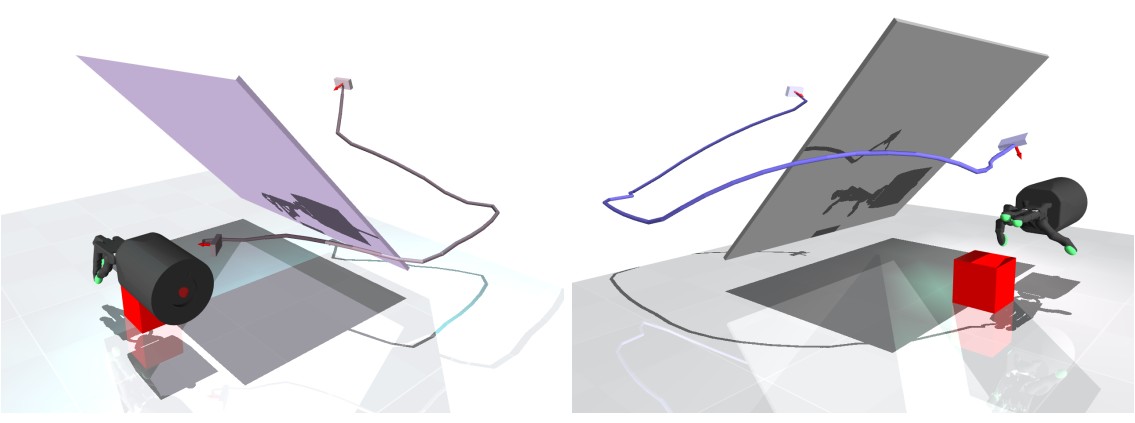

(a) Wrist flexion is viewed in profile.      (b) Wrist deviation is viewed from above.

Figure 9: Occlusion-avoidant trajectories, by the camera agent trained with random occlusions.

distribution. We randomize the lighting and expect that other standard data-randomization techniques could enable more robust sim2real transfers, as demonstrated in appendix E of Lee et al. (2022). At a higher level, this work does not address the open research question of how an agent can form an expanding conceptual space of internal GVF questions beyond predicting its own sensors.

**Future work** Giving our agent a foveal camera incentivizes it to position the subject more precisely at the center of the image. One direction for future work is to investigate whether the agent can maintain this centering as the target body part makes large movements, resulting in tracking behavior from the camera as a side-effect of framing. We demonstrate that a single agent can be trained to fixate on one of several targets, specified with a conditioning input. In this work, we limit the sensors to a single complex hand, where inferring joint angles of individual fingers can be challenging amidst the occlusions and visual distractions offered by the other fingers. We are therefore optimistic that our method would work on simpler morphologies such as parallel grippers, which we would like to explore in future work. A realistic body provides a rich variety of visual scales and distances, from shoulders to toe tips. We posit that these present a means to learn a rich repertoire of visual fixation and tracking skills without the need to design their reward functions. These skills could serve as a basis for exploration while learning higher-level skills, as demonstrated with SAC-X (Riedmiller et al., 2018), which used manually designed reward functions.

For example, some tasks are already known to benefit from fixating the eye on body parts as they touch the environment. Manipulation tasks such as pick-and-place benefit from a camera mounted on the wrist and fixated on the fingers (Hsu et al., 2022). Such tasks, however, are already well served by passive cameras. For future work, we envision active vision to primarily be of use in dynamic tasks with mobile robots, where the subject of visual interest can change rapidly.

Our learned self-prediction reward, while noisy, is shaped and temporally dense, which are attractive qualities for an auxiliary reward. Such rewards see extensive use where sparse task reward alone is insufficient to learn dynamic motor policies, such as playing football/soccer (Vezzani et al., 2022; Liu et al., 2022). We emphasize that the sensor prediction reward is not a means to observe one's own body parts *per se*, but rather a means to direct the gaze to whatever is conducive to predicting sensory inputs, such as the aural and tactile experience of making contact with a football. A reward for predicting contact with the ball incentivizes a robot to keep the ball in view, and is thus worth exploring as an auxiliary reward. Such downstream tasks exhibit different sensor distributions than the random policy used in this paper, and open interesting questions as to whether to fine-tune the predictor on the new distribution, or freeze it after pretraining on an entropy-maximizing policy.

Parisi et al. (2022) show that visual RL agents benefit from pretraining on static image datasets, even when the dataset imagery looks nothing like the tasks. Self-supervised RL presents a means of feature learning that is more natural, in a lifelong learning sense, than using human-annotated image datasets. Our method

learns features that are sufficient for the task of precise visual fixation. Whether they improve performance on general control tasks remains to be seen.

## 6 Conclusion

Learning motor skills without externally defined rewards or tasks is one path towards lifelong learning. Externally specified rewards can require privileged information that is unavailable to a real embodied agent. This is especially true of skills such as active vision, employed across a wide variety of embodied visual tasks.

In this paper, we demonstrate a means of learning visual fixation from self-prediction alone. We train a single embodied agent to visually fixate on different parts of its own body, as chosen by a conditioning input from a set of proprioceptive sensors. We show that this encourages the emergence of a *behavioral fovea*, where the fixated body part typically appears in a specialized region of the image, even when using conventional cameras. We show that when provided with an actual foveated camera, the same agent more strongly constrains the target to the center of the fovea.

We show that the agent learns to adopt distinct points of view for observing different sensor targets, fixating on them precisely, while circumnavigating occlusions. Taken together, these results present a means to learn a variety of visual skills, up to one per proprioceptive sensor, without the need for hand-designed rewards.

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
