# OpenReview forum: "Learning to Look by Self-Prediction"
_TMLR — Accepted by TMLR_

### Review · Reviewer_mMVG · 2022-12-20

**Summary Of Contributions:**

This paper presents an active vision framework for predicting proprioceptive senses. An RL agent learns to navigate to better viewpoints for predicting proprioceptive states of a target sensor (eg. the thumb joint). Unlike supervised methods, this agent is trained purely with TD losses from multi-modal alignment and doesn’t rely on any prior 3D information or demonstrations. The framework is conceptually simple, and also motivated by ideas from human/animal perception (like “behavioral fovea”). Experiments include evaluations in Mujoco with a simulated multi-fingered hand. These evaluations measure proprioceptive errors in joint angles or binary variables. Baselines include blind agents (that guess the most likely proprioceptive states), oracle agents (that used fixed cameras), and also an ablation investigating the usefulness of foveal over standard cameras. The proposed framework makes substantially lower prediction errors than all baselines.

**Audience:**

Yes

**Broader Impact Concerns:**

As such, the reviewer does not see any major concerns. Training such an agent in the real-world with RL is a safety concern, but the authors seem keen on taking the sim2real route, which should be safer.

**Claims And Evidence:**

Yes

**Requested Changes:**

The paper as is already relevant and interesting to the robotics community. The following changes could potentially improve incentives for broader adoption:
+ Show results on usefulness for downstream manipulation tasks.
+ Study the importance of low proprioceptive predictive error on success rates of downstream tasks.
+ Reproduce the results with real-robots.


**Strengths And Weaknesses:**

Strengths:
+ The overall “behavioral fovea” approach to active perception for robotics is quite novel and an interesting direction for the community. Quite often, methods use explicit 3D representations (like pointclouds, or voxels) or ignore 3D information entirely with RGB-only encoders. Multi-modal alignment is a general prior for robotics, and it’s great to see this paper tackle this area without making additional assumptions.
+ The evaluations cover a good range of baselines and ablations. This discussions accompanying the results do a good job of motivating the design choices and clearly state the benefits of the proposed approach over alternatives.
+ In general, the paper is well written and easy to understand. The video and figures are informative. Limitations are acknowledged.
+ If the authors decide to open-source the framework, or even just model, the robotics community could potentially extend it to other embodiments like legged robots and parallel-jaw manipulators.

Weaknesses
- While the paper focuses on predicting proprioception, it’s unclear how useful the framework would be for downstream manipulation tasks. Can the trained RL agent be directly used to find better viewpoints for BC or RL control policies, and if so, is it better than 1-3 fixed cameras (which is the common setup in robotics)? Or can the vision encoders be used as a pre-trained network for fine-tuning manipulation policies? In the natural world, prediction errors are closely tied with an agent’s need to achieve certain behaviors or goals other than just viewpoint correction. It would be great to explicitly investigate this.
- It’s also unclear how accurate proprioceptive predictions need to be in order to act capably. While the proposed approach outperforms all baselines in Table 1 in terms of joint angle errors, what is the best error rate that’s desirable? Clearly lower is better, but it’s unlikely that humans/animals make fine-grained predictions at the metric level. Perhaps a single, fixed, high-resolution camera is sufficient for “good-enough” prediction errors, but without knowing the underlying task this is hard to investigate.
- As acknowledged in the limitations, sim2real and/or real-world reproducibility is a concern. Domain randomization might be feasible in no-object scenes, but it could get complicated with one or multi-object scenes where interactions with the environment and objects adds another degree of complexity.  Further, the action-space of a free-floating camera is easy to simulate but hard to replicate in the real-world where collision avoidance and unreachable viewpoints have to be accounted for.

---

> ### Author Response · Authors · 2023-01-21
> **Responses to comments**
>
> We thank the reviewer for all their comments, and are heartened by their evaluation that “the paper as-is is already relevant and interesting to the robotics community.” We address the raised concerns below. We have uploaded a revised version of the paper, with additions highlighted in red.
>
> 1) How useful is the framework for downstream tasks?
>
> We agree that demonstrating benefits to downstream tasks would broaden interest. In the uploaded revision, we have added new text to the future work section (highlighted in red), discussing how some tasks, such as pick-and-place manipulation, are already known to benefit from having cameras continuously fixating on their end-effectors. We also add discussion on how, aside from skill transfer, the self-prediction reward is a candidate for consideration as a dense, shaped auxiliary reward to supplement sparse and binary task rewards, citing potential tasks.
>
> 2) Is active vision better than multiple fixed cameras?
>
> While multiple fixed cameras may improve on single fixed cameras, both are best suited for tasks limited to take place within their fixed field of view, such as manipulation within cages. For mobile robots such as quadrupeds, multiple cameras may be mounted to fixed positions on the robot. In some sense, this increases the burden of having to learn equivariant visual representations, by having to learn implicit mappings of 3D object position to their 2D positions on all cameras. We find that our active camera reduces this burden by fixating subjects at consistent locations in the retina (the behavioral fovea). In the future work section, we have added discussion on how our method might learn to observe external objects such as balls, in football/soccer. We agree that comparing against fixed cameras on such dynamic tasks would be of interest in future work.
>
> 3) How accurate do predictions need to be to act capably?
>
> We have added new plots to figure 4, and added a new section 4.3, which together show that even the noisy reward coming from an unconverged predictor is capable of training the camera agent to look at the subjects being predicted. The behavioral fovea histograms of figure 6c show that this fixation behavior is precise. The question of whether this noisy predictor is useful for downstream tasks in some sense reduces to whether the self-fixation behavior is useful to downstream tasks, which we argue in the affirmative in response to the reviewer’s question 1 above.
>
> 4) Sim-to-real, domain randomization, and deploying on a real robot.
>
> We agree that sim-to-real is nontrivial. We have performed additional experiments to show at least that the agent is robust to large amounts of simulated sensor noise (section 4.3). The base learning algorithms have been evaluated on real robots: GVFs in Rich Sutton et al 2011 (“Horde: A Scalable Real-time Architecture for Learning Knowledge from Unsupervised Sensorimotor Interaction Categories and Subject Descriptors”), and D4PG in Serkan Cabi et al 2019 (“Scaling data-driven robotics with reward sketching and batch reinforcement learning”).  An exciting direction for future work is to validate this approach on physical robots, as we mention in the Limitations section.
>
> The reviewer has correctly identified the challenge and complexity of sim-to-real in scenes involving interactions with multiple objects. We point out that sim-to-real via domain randomization has been successfully deployed in such settings, including manipulation with multiple objects. See for example appendix E of Lee et al 2021: (“Beyond Pick-and-Place:
> Tackling Robotic Stacking of Diverse Shapes”). We have added this reference to the Limitations section. We also agree that deploying on a real robot would serve as an existence proof of successful sim-to-real and increase the work’s impact, but it is out of scope for the current paper.  Results in simulations are commonly used to guide real robot experimentation, and are more easily replicable by other research groups who have different (or no) physical robots.  We expect that the experiments provided in this paper will be sufficient for other researchers to build on the algorithms developed here.
>
> 5) In the real world, the camera bot may collide and / or have unreachable viewpoints.
>
> We agree that this is another dimension of the increase in complexity to be dealt with upon deploying on a real robot. While the camera is mounted on a geometrically feasible armature (the “camera bot”), that arm indeed has nothing to collide with in our uncluttered setup. However, the camera itself does collide with objects, such as the occluders of figure 8, forcing the policy to learn to circumnavigate them, rather than simply punch through.

---

### Review · Reviewer_mHZx · 2022-12-21

**Summary Of Contributions:**

This paper introduces a method for active perception that encourages an agent to move the camera to observe a robot’s sensors without explicit labels on the sensors’ locations. Specifically, the authors focus on a dexterous hand and aim to train camera agents to look at the tactile sensors on the fingertips.

The authors propose to do this task by training a camera control agent that obtains visual observations, based on which a visual predictor network predicts the future returns of a specific sensor. Intuitively, the camera control agent wants to focus on regions that are helpful for the predictor network to predict sensing information.

Experimental results suggest that the agent moves the camera to points of view that are most predictive for a chosen sensor, even with learning-based noisy reward functions. The learned policies reliably place the sensor in a specific location in the view. By replacing the conventional camera with a foveal camera, the policies’ precision can be further increased.

**Audience:**

Yes

**Broader Impact Concerns:**

I do not see any concerns regarding ethical aspects of this work.

**Claims And Evidence:**

Yes

**Requested Changes:**

Besides the points in the weakness section, I hope to see the following two additional changes.

What will happen if the highest resolution region is not at the center of the foveal camera? Will the agent still place the sensor at the location with the highest resolution?

I’m expecting more discussions and connections to the biology findings, which I believe can further extend the paper’s impact to a broader community.

**Strengths And Weaknesses:**

[Strengths]

This is an interesting application of using a learning-based reward function (i.e., predicting the sensing information) to train an RL agent to actively perceive the environment.

The connection to biology, especially computational models of foveation, is also interesting and could be of relevance to a broader audience beyond the machine learning community.

The study of a foveal camera further verifies the effectiveness of the proposed algorithm, where the agent frames the sensor on the camera regions with the highest resolution.

[Weaknesses]

This paper only contains simulation results. It would greatly increase the paper’s significance if the authors conducted real-world experiments using real robots to control the cameras.

Related to my previous point, it is unclear how large the sim-to-real gap is using the current simulation setup. As a result, I’m not entirely convinced if the observations presented in this paper still hold true when extending to the real world.

This paper only considers touch as the sensing modality to make the prediction. It would greatly strengthen the paper if the authors could consider other sensing modalities, like sound (which might require the agent to focus more on the motion of the hand) and vision (which might require the camera agent to cover the field of view of the predicted vision sensors).

This paper currently also only considers a dexterous hand. Will the conclusion still hold true for other manipulators like parallel grippers or dexterous hands with other morphologies?

As has been mentioned in the limitation section, the hand is moving in a random fashion, which is very different from how humans would use their hands. Therefore, the sensing information distribution would be very different from a policy that can do meaningful tasks. Training the models in scenarios where the hand is doing practical tasks can improve the impact of this paper.

---

> ### Author Response · Authors · 2023-01-21
> **Response part 1**
>
> We thank the reviewer for their comments.  We appreciate the identification of strengths, and address the raised concerns in the points below. We have uploaded a revised version of the paper, with additions highlighted in red.
>
> 1) Off-center fovea?
>
> The reviewer asks if the foveal camera would continue to frame the target at the fovea, if the fovea is off-center. This is indeed an intriguing question, given that the non-foveal camera also frames targets at the center. We have now tested whether the foveal camera’s tighter central focus is because of its fovea, or merely an enhancement of a preexisting central bias to the learning dynamics. We have added a new figure 7, which runs our experiment with three different horizontal offsets of the physical fovea. It shows that indeed, each agent learns to fixate all sensor targets on the physical fovea, even when the fovea is offset from center.
>
> 2) The sim-to-real gap
>
> The authors agree with the reviewer that real robot evaluation is an important step for future work. To examine our method’s robustness, we have added experiments that add large amounts of noise to the target sensors (Figure 5).  Moreover, the base learning algorithms have been evaluated on real robots: GVFs in Rich Sutton et al 2011 (“Horde: A Scalable Real-time Architecture for Learning Knowledge from Unsupervised Sensorimotor Interaction Categories and Subject Descriptors”), and D4PG in Serkan Cabi et al 2019 (“Scaling data-driven robotics with reward sketching and batch reinforcement learning”).
>
> 3) The paper would be stronger with experiments on real robots.
>
> The authors agree with the merits of performing experiments on real robots, but it is out of scope for the current paper.  Results in simulations are commonly used to guide real robot experimentation, and are more easily replicable by other research groups who have different (or no) physical robots.  We expect that the experiments provided in this paper will be sufficient for other researchers to build on the algorithms developed here.
>
> 4) The paper only considers touch as a target, what about sound and vision?
>
> Thank you for the interesting suggestion of using sound and vision as targets. We would briefly correct the above to say this paper considers many sensors other than touch, although they are all proprioceptive sensors.  Sound is an interesting challenge, but it is not easily modeled in most simulators. Simple acoustic impulses are nonetheless tempting to predict, as they may be valuable for predicting e.g. impact with a ball. We have added discussion of this in the Future Work section. Vision is a more accessible target for prediction, and this raises an interesting question: Can the proposed algorithm work for improving visual target prediction if we flipped the role of the camera and the hand (so if the camera motion were driven by Perlin noise, the prediction targets were pixels from the image, only the hand could be controlled to influence the visual image).   This is an interesting thought experiment to see how far these ideas could go, and it would be interesting to pursue at a later stage.  We have outlined some practical next experiments in the future work section.
>
> 5) Testing with Parallel Grippers and other morphologies?
>
> We agree with the reviewer that testing this approach with other bodies is valuable.  While we already suggest using full bodies in future work, we have now extended this discussion to include different manipulator morphologies including parallel grippers, as well as dynamic full-bodied tasks.  Adding such experiments to this paper is out of scope, as making a good simulation of new robot morphologies is a substantial effort.

---

> ### Author Response · Authors · 2023-01-21
> **Response part 2**
>
> 6) Practical task motion instead of randomly-driven motion?
>
> We agree that the random policy driving the hand will exhibit different sensor statistics than any given task-specific behavior. As noted by the reviewer, the Limitations section points out that the random hand policy causes the touch sensor to fire only rarely, a label imbalance that makes touch prediction more difficult to learn than it might otherwise be in a touch-heavy manipulation task. Conversely, other sensors may be more difficult to target under a non-random policy. We have added discussion of this, and potential means of mitigating it, to the Future Work section.  The main purpose of this work is to see whether goal-directed behavioral skills can be acquired in the absence of any task direction.  As mentioned in the Future Work section, methods such as SAC-X (Reidmiller et al, 2018) rely on such auxiliary tasks to provide guidance towards task-driven performance.  Moreover, a motivating theme in modern ML (and reinforcement learning), is to stumble into the exploitable regularities in a vast quantity  of noisy experience.  We expect that this overall approach would continue to work within an experience-stream of task-directed behavior (in the same manner that policy improvement is generally effective in reinforcement learning).
>
> 7) Connections to biology?
>
> Thank you for the suggestion. To further motivate the study of active vision, we have added animal examples of active vision to the previously human-centered discussion in the introduction. This is in addition to the discussion of behavioral fovea in rats, in the Discussion section.

---

### Review · Reviewer_WqZV · 2022-12-24

**Summary Of Contributions:**

This paper presents a framework to learn visual servoing policies without specifying an explicit reward function. It trains a GVF that predicts discounted, distributional cumulative sensor readings. Policies are trained with D4PG by maximizing the negative GVF prediction error with the selected joint. Experiments show that 1. The policy learns emergent fovea behavior and 2. The policy learns emergent occlusion circumventing behavior.


**Audience:**

Yes

**Broader Impact Concerns:**

No ethical concerns.

**Claims And Evidence:**

Yes

**Requested Changes:**

Also as mentioned above
1. More discussions on how the presented framework handle the aleatoric and epistemic uncertainty of the underlying system.
2. More discussions on how the presented framework work in a more complicated manipulation setting.
3. Comparison with a classical policy that uses the robot’s geometry, as mentioned in 4.4.


**Strengths And Weaknesses:**

### Strengths
+ The paper is well written, motivated and presented. Experiments are complete.
+ Insightful qualitative analysis on the emergent behavior.

### Weaknesses
- The intrinsic reward formulation with the prediction errors might be susceptible to sensor reading noise, which is common in the real world. Would the policies learn to simply posit cameras to regions that are less susceptible to sensor noise as opposed to learning a useful behavior? Concretely, I would like to see more discussions on how this formulation can handle the aleatoric and epistemic uncertainty of the underlying system.
- I would like to see how the method works in a more complex setting, for example, pick and place. When the camera is free to look at regions so as to better perform the manipulation task, what behavior would be the policy learned under that setting? Overall, I would like to see more discussions on how such a framework could be used for downstream robotics tasks.
- Comparison with a classical policy that uses the robot’s geometry, as mentioned in 4.4.

---

> ### Author Response · Authors · 2023-01-21
> **Response part 1: Aleatoric uncertainty**
>
> We thank the reviewer for their assessment of our qualitative analysis, and are particularly glad that they found our experiments to be complete. Nevertheless, we took the liberty of performing additional experiments to address their questions, as below. We have uploaded a revised version of the paper, with additions highlighted in red.
>
> 1) Sensor noise and uncertainty
>
> The reviewer rightly asks how our method fares in the face of uncertain predictions, given that it uses prediction accuracy as a reward. Predictions may be uncertain due to external, aleatoric sources of uncertainty, such as sensor noise, or internal, epistemic uncertainty, such as a sub-optimal predictor within the function space. To address each, we have added a new section 4.3, and a new figure 5. These are highlighted in red in the revised PDF, and while we encourage the reviewer to have a look, we have summarized the changes below.
>
> 1.1) Aleatoric uncertainty
>
> Although sensor noise will worsen predictions, it remains unlikely that looking away from the target sensor entirely will improve the predictions. We posit that it would take a pathological amount of sensor noise for this to happen, at which point the robot would have even bigger problems than learning to look.
>
> To test this, we ran a sweep of experiments with Gaussian noise added to all the non-visual sensors. We swept the hyperparameter $\alpha$ that defines the standard deviation of the normal distribution, and from which we sample sensor noise: $\sigma = \alpha * (sensor\_max\_value - sensor\_min\_value) / 2$. In other words, if $\alpha = 1$, the range of the noise distribution’s $\pm 1$ standard deviation is as wide as the sensor range itself. We try values of $\alpha$ up to 0.4, an amount of noise that would be considered extreme in a robotic context.
>
> We find that the camera agent remains able to improve the prediction error even under such conditions. We have added figure 5 to the paper, showing the training curves of agent performance under various amounts of sensor noise. For reference, we have compiled the final values of the curves to the table below. Each row shows its noise amount (alpha), and the camera agent performance for each sensor target. The numbers are the amount by which the agent reduced the prediction error over the course of an episode, by moving the camera. Bigger is better, with any positive value indicating that the camera agent has moved the camera to a view with better prediction error.
>
> | alpha | Forefinger root joint | Thumb root joint | Wrist flexion | Wrist deviation |
> |-------|-----------------------|------------------|---------------|-----------------|
> | 0.0   | 0.61                  | 0.79             | 0.51          | 0.66            |
> | 0.05  | 0.69                  | 1.01             | 0.48          | 0.70            |
> | 0.1   | 0.52                  | 0.78             | 0.49          | 0.64            |
> | 0.2   | 0.42                  | 0.66             | 0.38          | 0.65            |
> | 0.4   | 0.21                  | 0.39             | 0.16          | 0.52            |

---

> ### Author Response · Authors · 2023-01-21
> **Response part 2: Epistemic uncertainty**
>
> 1.2) Epistemic uncertainty
>
> Although our agent is not a bayesian agent that maintains an explicit representation of epistemic uncertainty, we can still examine its robustness to epistemic uncertainty if we interpret “epistemically uncertain” in the general sense of “having an imperfect predictor” relative to what the agent can represent.  In particular, we can compare performance measures at earlier and later stages of training. To this end, we have added an extra row of plots below figure 4. The old, first row shows training curves that represent agent competence, as partially separated from predictor competence. These plot the improvement in target sensor prediction error between the start and end of an episode, due to camera movement. The second row shows corresponding training curves showing predictor competence as the target predictor error averaged over an episode. These parallel curves show that the agent enters the plateau of its performance (first row) well in advance of the predictor converging to its minimum error (second row). This indicates that the agent learns to move the camera to improve predictions, even when the predictor’s epistemic uncertainty (reducible error) is still far from optimal. Below we summarize figure 4’s training curves in table form. For each target sensor, it shows the training time T needed for the agent to reach 50% of its final performance, and the percent convergence of the predictor at that time. It shows that the policies for the more visually salient joints (thumb, wrist) reach 50% performance well in advance of the predictor doing so.
>
> |                                  | little finger touch | Forefinger root joint | Thumb root joint | Wrist flexion | Wrist deviation |
> |----------------------------------|---------------------|-----------------------|------------------|---------------|-----------------|
> | T: time to 50% actor convergence | 27500 s             | 28000 s               | 7200 s           | 6000 s        | 5500 s          |
> | predictor convergence at t=T     | 96%                 | 77%                   | 11%              | 36%           | 22%             |

---

> ### Author Response · Authors · 2023-01-21
> **Response part 3: discussion of downstream tasks, comparison with a classical policy**
>
> 2) Discussion of downstream tasks
>
> We have added discussion in the future work section about specific downstream tasks, including the pick-and-place task mentioned by the reviewer. In addition to the specific tasks, we discuss in general how these active vision skills could transfer to downstream tasks, and argue that our prediction reward could augment sparse task rewards as a dense and shaped auxiliary reward.
>
> 3) Comparison with a classical policy that uses the robot’s geometry, as mentioned in 4.4
>
> We have performed this comparison and added it as additional rows to table 1, our table of comparison baselines. The new baseline looks at the target sensor using its position as computed from the robot’s own geometry. The table shows that such a geometric policy outperforms the random agent, but underperforms ours. We have added discussion of this baseline to both section 4.2.1, which describes it in detail, and 4.5 (previously 4.4), which originally brought up the geometric policy.

---

### Decision · Action_Editors · 2023-02-02

**Recommendation:** Accept as is

**Comment:**

Reviewers posed a number of questions about uncertainty, robustness, and how these techniques may transfer to real robotic applications or be extended beyond the sensors used. Authors addressed some of these, but the others remain as interesting follow-up work.

**Audience:**

Reviewers agree that there is likely an interested audience within the TMLR community. The AE agrees.

**Claims And Evidence:**

Reviewers unanimously agree that the claims in this manuscript are well-supported and have been further strengthened by the updated sensor-noise experiments. The AE agrees.